# Synthetic Experience Replay

**Cong Lu**[*]**, Philip J. Ball**[*]**, Yee Whye Teh, Jack Parker-Holder**
University of Oxford

## Abstract

A key theme in the past decade has been that when large neural networks and large datasets combine they can produce remarkable results. In deep reinforcement learning (RL), this paradigm is commonly made possible through *experience replay*, whereby a dataset of past experiences is used to train a policy or value function. However, unlike in supervised or self-supervised learning, an RL agent has to collect its own data, which is often limited. Thus, it is challenging to reap the benefits of deep learning, and even small neural networks can overfit at the start of training. In this work, we leverage the tremendous recent progress in generative modeling and propose Synthetic Experience Replay (SYNTHER), a diffusion-based approach to flexibly upsample an agent's collected experience. We show that SYNTHER is an effective method for training RL agents across offline and online settings, in both proprioceptive and pixel-based environments. In offline settings, we observe drastic improvements when upsampling small offline datasets and see that additional synthetic data also allows us to effectively train larger networks. Furthermore, SYNTHER enables online agents to train with a much higher update-to-data ratio than before, leading to a significant increase in sample efficiency, *without any algorithmic changes*. We believe that synthetic training data could open the door to realizing the full potential of deep learning for replay-based RL algorithms from limited data. Finally, we open-source our code at `https://github.com/conglu1997/SynthER`.

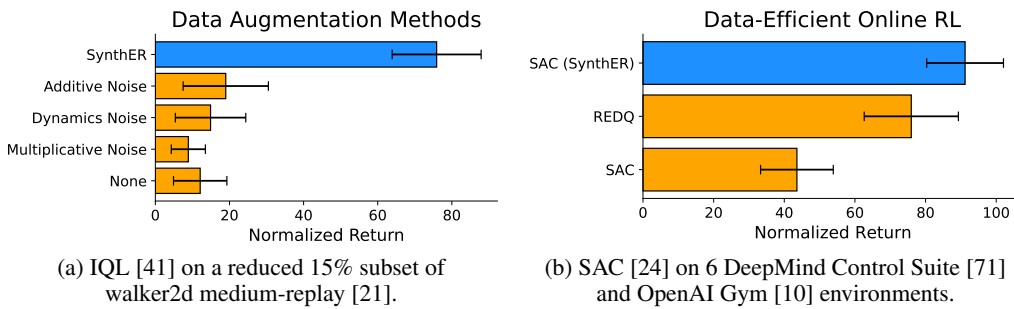

(a) IQL [41] on a reduced 15% subset of walker2d medium-replay [21].

(b) SAC [24] on 6 DeepMind Control Suite [71] and OpenAI Gym [10] environments.

Figure 1: Upsampling data using SYNTHER greatly outperforms explicit data augmentation schemes for small offline datasets and data-efficient algorithms in online RL *without any algorithmic changes*. Moreover, synthetic data from SYNTHER may readily be added to *any* algorithm utilizing experience replay. Full results in Section 4.

## 1 Introduction

In the past decade, the combination of large datasets [14, 63] and ever deeper neural networks [15, 25, 43, 73] has led to a series of more generally capable models [11, 56, 58]. In reinforcement learning (RL, Sutton and Barto [67]), agents typically learn online from their own experience. Thus,

---

[*]Equal contribution. Correspondence to `cong.lu@stats.ox.ac.uk` and `ball@robots.ox.ac.uk`.

37th Conference on Neural Information Processing Systems (NeurIPS 2023).

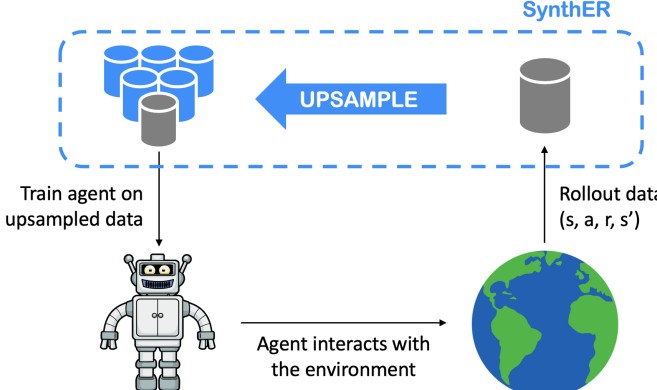

Figure 2: SYNTHER allows any RL agent using experience replay to arbitrarily upsample, or increase the quantity of, their experiences (in grey) and train on synthetic data (in blue). We evaluate our approach up to a factor of $100\times$ more data in Section 4.2, across both proprioceptive and pixel-based environments. By leveraging this increased data, agents can learn effectively from smaller datasets and achieve higher sample efficiency. Details of the upsampling process are given in Figure 3.

to leverage sufficiently rich datasets, RL agents typically make use of *experience replay* [19, 53], where training takes place on a dataset of recent experiences. However, this experience is typically limited, unless an agent is distributed over many workers which requires both high computational cost and sufficiently fast simulation [18, 37].

Another approach for leveraging broad datasets for training RL policies is *offline* RL [2, 46], whereby behaviors may be distilled from previously collected data either via behavior cloning [61], off-policy learning [22, 44] or model-based methods [39, 49, 78]. Offline data can also significantly bootstrap online learning [9, 27, 75]; however, it is a challenge to apply these methods when there is a mismatch between offline data and online environment. Thus, many of the successes rely on toy domains with transfer from specific behaviors in a simple low-dimensional proprioceptive environment.

Whilst strong results have been observed in re-using prior data in RL, appropriate data for particular behaviors may simply not exist and thus this approach falls short in generality. We consider an alternative approach—rather than passively reusing data, we leverage tremendous progress in generative modeling to generate a large quantity of new, synthetic data. While prior work has considered upsampling online RL data with VAEs or GANs [32, 34, 51], we propose making use of *diffusion* generative models [30, 38, 66], which unlocks significant new capabilities.

Our approach, which we call *Synthetic Experience Replay*, or SYNTHER, is conceptually simple, whereby given a limited initial dataset, we can arbitrarily upsample the data for an agent to use as if it was real experience. Therefore, in this paper, we seek to answer a simple question: *Can the latest generative models replace or augment traditional datasets in reinforcement learning?* To answer this, we consider the following settings: offline RL where we entirely replace the original data with data produced by a generative model, and online RL where we upsample experiences to broaden the training data available to the agent. In both cases, SYNTHER leads to drastic improvements, obtaining performance comparable to that of agents trained with substantially more real data. Furthermore, in certain offline settings, synthetic data enables effective training of larger policy and value networks, resulting in higher performance by alleviating the representational bottleneck. Finally, we show that SYNTHER scales to pixel-based environments *by generating data in latent space*. We thus believe this paper presents sufficient evidence that our approach could enable entirely new, efficient, and scalable training strategies for RL agents. To summarize, the contributions of this paper are:

- We propose SYNTHER in Section 3, a diffusion-based approach that allows one to generate synthetic experiences and thus arbitrarily upsample data for any reinforcement learning algorithm utilizing experience replay.

- We validate the synthetic data generated by SYNTHER in offline settings across proprioceptive and pixel-based environments in Section 4.1 and Section 4.3, presenting the first generative approach to show parity with real data on the standard D4RL and V-D4RL offline datasets with a wide variety of algorithms. Furthermore, we observe considerable improvements from upsampling for small offline datasets and scaling up network sizes.

- We show how SYNTHER can arbitrarily upsample an online agent's training data in Section 4.2 by continually training the diffusion model. This allows us to significantly increase an agent's update-to-data (UTD) ratio matching the efficiency of specially designed data-efficient algorithms *without any algorithmic changes*.

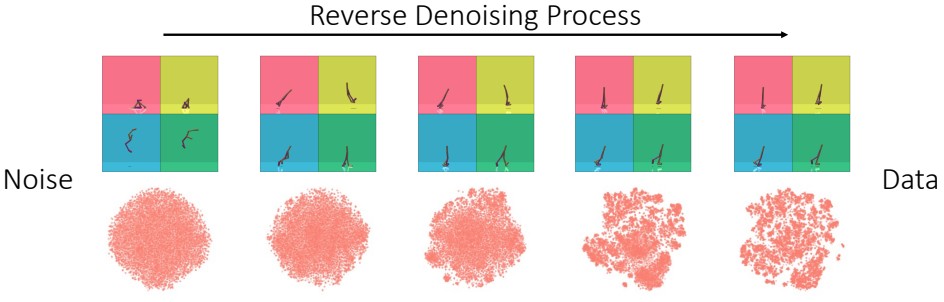

Figure 3: SYNTHER generates synthetic samples using a diffusion model which we visualize on the proprioceptive walker2d environment. On the **top row**, we render the state component of the transition tuple on a subset of samples; and on the **bottom row**, we visualize a t-SNE [72] projection of 100,000 samples. The denoising process creates cohesive and plausible transitions whilst also remaining diverse, as seen by the multiple clusters that form at the end of the process in the bottom row.

## 2 Background

### 2.1 Reinforcement Learning

We model the environment as a Markov Decision Process (MDP, Sutton and Barto [67]), defined as a tuple $M = (\mathcal{S}, \mathcal{A}, P, R, \rho_0, \gamma)$, where $\mathcal{S}$ and $\mathcal{A}$ denote the state and action spaces respectively, $P(s'|s,a)$ the transition dynamics, $R(s,a)$ the reward function, $\rho_0$ the initial state distribution, and $\gamma \in (0,1)$ the discount factor. The goal in reinforcement learning is to optimize a policy $\pi(a|s)$ that maximizes the expected discounted return $\mathbb{E}_{\pi,P,\rho_0} \left[ \sum_{t=0}^{\infty} \gamma^t R(s_t, a_t) \right]$.

### 2.2 Offline Reinforcement Learning

In *offline RL* [46], the policy is not deployed in the environment until test time. Instead, the algorithm only has access to a static dataset $\mathcal{D}_{\text{env}} = \{(s_t, a_t, r_t, s_{t+1})\}_{t=1}^{T}$, collected by one or more behavioral policies $\pi_b$. We refer to the distribution from which $\mathcal{D}_{\text{env}}$ was sampled as the *behavioral distribution* [78]. In some of the environments we consider, the environment may be finite horizon or have early termination. In that case, the transition tuple also contains a terminal flag $d_t$ where $d_t = 1$ indicates the episode ended early at timestep $t$ and $d_t = 0$ otherwise.

### 2.3 Diffusion Models

Diffusion models [30, 66] are a class of generative models inspired by non-equilibrium thermodynamics that learn to iteratively reverse a forward noising process and generate samples from noise. Given a data distribution $p(\mathbf{x})$ with standard deviation $\sigma_{\text{data}}$, we consider noised distributions $p(\mathbf{x}; \sigma)$ obtained by adding i.i.d. Gaussian noise of standard deviation $\sigma$ to the base distribution. The forward noising process is defined by a sequence of noised distributions following a fixed noise schedule $\sigma_0 = \sigma_{\text{max}} > \sigma_1 > \cdots > \sigma_N = 0$. When $\sigma_{\text{max}} \gg \sigma_{\text{data}}$, the final noised distribution $p(\mathbf{x}; \sigma_{\text{max}})$ is essentially indistinguishable from random noise.

Karras et al. [38] consider a probability-flow ODE with the corresponding continuous noise schedule $\sigma(t)$ that maintains the desired distribution as $\mathbf{x}$ evolves through time given by Equation (1).

$$\mathrm{d}\mathbf{x} = -\dot{\sigma}(t)\sigma(t)\nabla_{\mathbf{x}} \log p(\mathbf{x}; \sigma(t))\mathrm{d}t \tag{1}$$

where the dot indicates a time derivative and $\nabla_{\mathbf{x}} \log p(\mathbf{x}; \sigma(t))$ is the score function [33], which points towards the data at a given noise level. Infinitesimal forward or backward steps of this ODE either nudge a sample away or towards the data. Karras et al. [38] consider training a denoiser $D_\theta(\mathbf{x}; \sigma)$ on an L2 denoising objective:

$$\min_{\theta} \mathbb{E}_{\mathbf{x} \sim p, \sigma, \epsilon \sim \mathcal{N}(0, \sigma^2 I)} \|D_\theta(\mathbf{x} + \epsilon; \sigma) - \mathbf{x}\|_2^2 \tag{2}$$

and then use the connection between score-matching and denoising [74] to obtain $\nabla_{\mathbf{x}} \log p(\mathbf{x}; \sigma) = (D_\theta(\mathbf{x}; \sigma) - \mathbf{x})/\sigma^2$. We may then apply an ODE (or SDE as a generalization of Equation (1)) solver to reverse the forward process. In this paper, we train our diffusion models to approximate the online or offline behavioral distribution.

---

**Algorithm 1** SYNTHER for online replay-based algorithms. Our additions are highlighted in **blue**.

1: **Input:** real data ratio $r \in [0, 1]$
2: **Initialize:** $\mathcal{D}_{\text{real}} = \emptyset$ real replay buffer, $\pi$ agent, $\mathcal{D}_{\text{synthetic}} = \emptyset$ synthetic replay buffer, $M$ diffusion model
3: **for** $t = 1, \ldots, T$ **do**
4:     Collect data with $\pi$ in the environment and add them to $\mathcal{D}_{\text{real}}$
5:     Update diffusion model $M$ with samples from $\mathcal{D}_{\text{real}}$
6:     Generate samples from $M$ and add them to $\mathcal{D}_{\text{synthetic}}$
7:     Train $\pi$ on samples from $\mathcal{D}_{\text{real}} \cup \mathcal{D}_{\text{synthetic}}$ mixed with ratio $r$
8: **end for**

---

## 3 Synthetic Experience Replay

In this section, we introduce Synthetic Experience Replay (SYNTHER), our approach to upsampling an agent's experience using diffusion. We begin by describing the simpler process used for offline RL and then how that may be adapted to the online setting by continually training the diffusion model.

### 3.1 Offline SYNTHER

For offline reinforcement learning, we take the data distribution of the diffusion model $p(\mathbf{x})$ to simply be the offline behavioral distribution. In the proprioceptive environments we consider, the full transition is low-dimensional compared with typical pixel-based diffusion. Therefore, the network architecture is an important design choice; and similarly to Pearce et al. [55] we find it important to use a residual MLP denoising [70] network. Furthermore, the choice of the Karras et al. [38] sampler allows us to use a low number of diffusion steps ($n = 128$) resulting in high sampling speed. Full details for both are provided in Appendix B. We visualize the denoising process on a representative D4RL [21] offline dataset, in Figure 3. We further validate our diffusion model on the D4RL datasets in Figure 8 in Appendix A by showing that the synthetic data closely matches the original data when comparing the marginal distribution over each dimension. In Section 4.3, we show the same model may be used for pixel-based environments by generating data in a low-dimensional latent space.

Next, we conduct a quantitative analysis and show that **the quality of the samples from the diffusion model is significantly better** than with prior generative models such as VAEs [40] and GANs [23]. We consider the state-of-the-art Tabular VAE (TVAE) and Conditional Tabular GAN (CTGAN) models proposed by Xu et al. [76], and tune them on the D4RL halfcheetah medium-replay dataset. Full hyperparameters are given in Appendix B.1. As proposed in Patki et al. [54], we compare the following two high-level statistics: **(1) Marginal:** Mean Kolmogorov-Smirnov [52] statistic, measuring the maximum distance between empirical cumulative distribution functions, for each dimension of the synthetic and real data; and **(2) Correlation:** Mean Correlation Similarity, measuring the difference in pairwise Pearson rank correlations [20] between the synthetic and real data.

We also assess downstream offline RL performance using the synthetic data with two state-of-the-art offline RL algorithms, TD3+BC [22] and IQL [41], in Table 1. The full evaluation protocol is described in Section 4.1. The diffusion model is far more faithful to the original data than prior generative models which leads to substantially higher returns on both algorithms. Thus, we hypothesize a large part of the failure of prior methods [34, 51] is due to a weaker generative model.

### 3.2 Online SYNTHER

SYNTHER may be used to upsample an online agent's experiences by continually training the diffusion model on new experiences. We provide pseudocode for how to incorporate SYNTHER

Table 1: SYNTHER is better at capturing both the high-level statistics of the dataset (halfcheetah medium-replay) than prior generative models and also leads to far higher downstream performance. Metrics (left) computed from 100K samples from each model, offline RL performance (right) computed using 5M samples from each model. We show the mean and standard deviation of the final performance averaged over 8 seeds.

| Model | Metrics | | Eval. Return | |
|---|---|---|---|---|
| | **Marginal** | **Correlation** | **TD3+BC** | **IQL** |
| Diffusion (Ours) | **0.989** | **0.998** | **45.9±0.9** | **46.6±0.2** |
| VAE [76] | 0.942 | 0.979 | 27.1±2.1 | 15.2±2.2 |
| GAN [76] | 0.959 | 0.981 | 24.3±1.9 | 15.9±2.4 |

Table 2: A comprehensive evaluation of SYNTHER on a wide variety of proprioceptive D4RL [21] datasets and selection of state-of-the-art offline RL algorithms. We show that synthetic data from SYNTHER faithfully reproduces the original performance, which allows us to completely eschew the original training data. We show the mean and standard deviation of the final performance averaged over 8 seeds. **Highlighted** figures show at least parity over each group (algorithm and environment class) of results.

| Environment | Behavioral Policy | TD3+BC [22] | | IQL [41] | | EDAC [5] | |
|---|---|---|---|---|---|---|---|
| | | Original | SYNTHER | Original | SYNTHER | Original | SYNTHER |
| halfcheetah-v2 | random | 11.3±0.8 | 12.2±1.1 | 15.2±1.2 | 17.2±3.4 | - | |
| | mixed | 44.8±0.7 | 45.9±0.9 | 43.5±0.4 | 46.6±0.2 | 62.1±1.3 | 63.0±1.3 |
| | medium | 48.1±0.2 | 49.9±1.2 | 48.3±0.1 | 49.6±0.3 | 67.7±1.2 | 65.1±1.3 |
| | medexp | 90.8±7.0 | 87.2±11.1 | 94.6±0.2 | 93.3±2.6 | 104.8±0.7 | 94.1±10.1 |
| walker2d-v2 | random | 0.6±0.3 | 2.3±1.9 | 4.1±0.8 | 4.2±0.3 | - | |
| | mixed | 85.6±4.6 | 90.5±4.3 | 82.6±8.0 | 83.3±5.9 | 87.1±3.2 | 89.8±1.5 |
| | medium | 82.7±5.5 | 84.8±1.4 | 84.0±5.4 | 84.7±5.5 | 93.4±1.6 | 93.4±2.4 |
| | medexp | 110.0±0.4 | 110.2±0.5 | 111.7±0.6 | 111.4±0.7 | 114.8±0.9 | 114.7±1.2 |
| hopper-v2 | random | 8.6±0.3 | 14.6±9.4 | 7.2±0.2 | 7.7±0.1 | - | |
| | mixed | 64.4±24.8 | 53.4±15.5 | 84.6±13.5 | 103.2±0.4 | 99.7±0.9 | 101.4±0.8 |
| | medium | 60.4±4.0 | 63.4±4.2 | 62.8±6.0 | 72.0±4.5 | 101.7±0.3 | 102.4±0.5 |
| | medexp | 101.1±10.5 | 105.4±9.7 | 106.2±6.1 | 90.8±17.9 | 105.2±11.6 | 109.7±0.2 |
| **locomotion average** | | **59.0±4.9** | **60.0±5.1** | **62.1±3.5** | **63.7±3.5** | **92.9±2.4** | **92.6±2.1** |
| maze2d-v1 | umaze | 29.4±14.2 | 37.6±14.4 | 37.7±2.0 | 41.0±0.7 | 95.3±7.4 | 99.1±18.6 |
| | medium | 59.5±41.9 | 65.2±36.1 | 35.5±1.0 | 35.1±2.6 | 57.0±4.0 | 66.4±10.9 |
| | large | 97.1±29.3 | 92.5±38.5 | 49.6±22.0 | 60.8±5.3 | 95.6±26.5 | 143.3±21.7 |
| **maze average** | | **62.0±28.2** | **65.1±29.7** | **40.9±8.3** | **45.6±2.9** | **82.6±12.6** | **102.9±17.1** |

into any online replay-based RL agent in Algorithm 1 and visualize this in Figure 2. Concretely, a diffusion model is periodically updated on the real transitions and then used to populate a second synthetic buffer. The agent may then be trained on a mixture of real and synthetic data sampled with ratio $r$. For the results in Section 4.2, we simply set $r = 0.5$ following Ball et al. [9]. The synthetic replay buffer may also be configured with a finite capacity to prevent overly stale data.

## 4 Empirical Evaluation

We evaluate SYNTHER across a wide variety of offline and online settings. First, we validate our approach on offline RL, where we entirely replace the original data, and further show large benefits from upsampling small offline datasets. Next, we show that SYNTHER leads to large improvements in sample efficiency in online RL, exceeding specially designed data-efficient approaches. Furthermore, we show that SYNTHER scales to pixel-based environments by generating data in latent space. Finally, we perform a meta-analysis over our empirical evaluation using the RLiable [3] framework in Figure 7.

### 4.1 Offline Evaluation

We first verify that synthetic samples from SYNTHER faithfully model the underlying distribution from the canonical offline D4RL [21] datasets. To do this, we evaluate SYNTHER in combination with 3 widely-used SOTA offline RL algorithms: TD3+BC (Fujimoto and Gu [22], explicit policy regularization), IQL (Kostrikov et al. [41], expectile regression), and EDAC (An et al. [5], uncertainty-based regularization) on an extensive selection of D4RL datasets. We consider the MuJoCo [69] locomotion (halfcheetah, walker2d, and hopper) and maze2d environments. In these experiments, all datasets share the same training hyperparameters in Appendix B, with some larger datasets using a wider network. For each dataset, we upsample the original dataset to **5M samples**; we justify this choice in Appendix C.1. We show the final performance in Table 2.

Our results show that we achieve at least parity for all groups of environments and algorithms as highlighted in the table, *regardless of the precise details of each algorithm*. We note significant improvements to maze2d environments, which are close to the 'best' performance as reported in CORL [68] (i.e., the best iteration during offline training) rather than the final performance. We hypothesize this improvement is largely due to increased data from SYNTHER, which leads to less overfitting and increased stability. For the locomotion datasets, we largely reproduce the original results, which we attribute to the fact that most D4RL datasets are at least 1M in size and are already sufficiently large. However, as detailed in Table 5 in Appendix A.1, SYNTHER allows the effective size of the dataset to be compressed significantly, up to $12.9\times$ on some datasets. Finally, we present results on the AntMaze environment in Appendix E.1, and experiments showing that the synthetic and real data are compatible with each other in Appendix E.2.

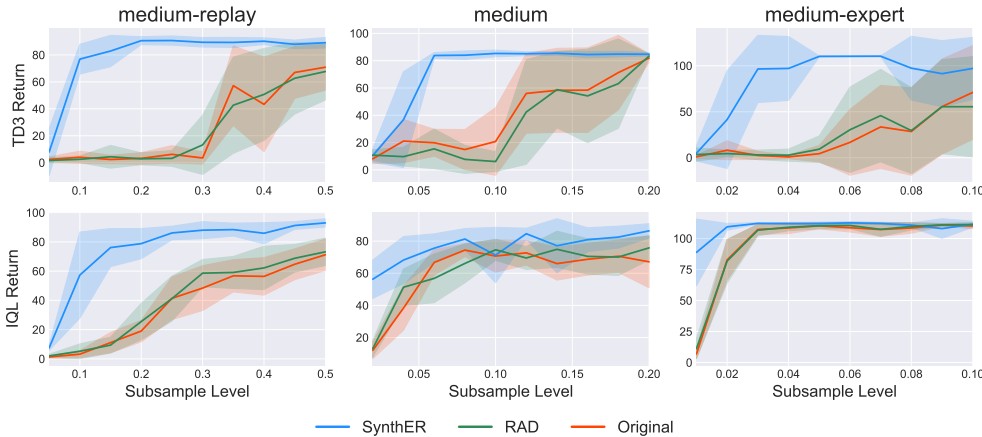

Figure 4: SYNTHER is a powerful method for upsampling reduced variants of the walker2d datasets and vastly improves on competitive explicit data augmentation approaches for both the TD3+BC (top) and IQL (bottom) algorithms. The subsampling levels are scaled proportionally to the original size of each dataset. We show the mean and standard deviation of the final performance averaged over 8 seeds.

### 4.1.1 Upsampling for Small Datasets

We investigate the benefit of SYNTHER for small offline datasets and compare it to canonical 'explicit' data augmentation approaches [8, 45]. Concretely, we wish to understand whether SYNTHER generalizes and generates synthetic samples that improve policy learning compared with *explicitly* augmenting the data with hand-designed inductive biases. We focus on the walker2d (medium, medium-replay/mixed, medium-expert) datasets in D4RL and uniformly subsample each at the transition level. We subsample each dataset proportional to the original dataset size so that the subsampled datasets approximately range from 20K to 200K samples. As in Section 4.1, we then use SYNTHER to *upsample* each dataset to 5M transitions. Our denoising network uses the same hyperparameters as for the original evaluation in Section 4.1.

In Figure 4, we can see that for all datasets, SYNTHER leads to a significant gain in performance and vastly improves on explicit data augmentation approaches. For explicit data augmentation, we select the overall most effective augmentation scheme from Laskin et al. [45] (adding Gaussian noise of the form $\epsilon \sim \mathcal{N}(0, 0.1)$). Notably, with SYNTHER we can achieve close to the original levels of performance on the walker2d-medium-expert datasets starting from **only 3% of the original data**. In Figure 1a, we methodically compare across both additive and multiplicative versions of RAD, as well as dynamics augmentation [8] on the 15% reduced walker medium-replay dataset.

**Why is SYNTHER better than explicit augmentation?** To provide intuition into the efficacy of SYNTHER over canonical explicit augmentation approaches, we compare the data generated by SYNTHER to that generated by the best-performing data augmentation approach in Figure 1a, namely additive noise. We wish to evaluate two properties: 1) How diverse is the data? 2) How accurate is the data for the purposes of learning policies? To measure diversity, we measure the *minimum* L2 distance of each datapoint from the dataset, which allows us to see how far the upsampled data is from the original data. To measure the validity of the data, we follow Lu et al. [49] and measure the MSE between the reward and next state proposed by SYNTHER with the true next state and reward defined by the simulator. We plot both these values in a joint scatter plot to compare how they vary with respect to each other. For this, we compare specifically on the reduced 15%

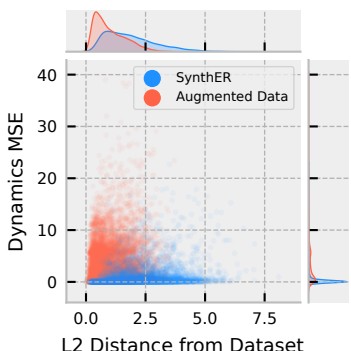

Figure 5: Comparing L2 distance from training data and dynamics accuracy under SYNTHER and augmentations.

subset of walker2d medium-replay as in Figure 1a. As we see in Figure 5, SYNTHER generates a significantly wider marginal distribution over the distance from the dataset, and generally produces samples that are further away from the dataset than explicit augmentations. Remarkably, however, we see that these samples are far more consistent with the true environment dynamics. Thus, SYNTHER

Table 3: SYNTHER enables effective training of larger policy and value networks for TD3+BC [22] leading to a **11.7%** gain on the offline MuJoCo locomotion datasets. In comparison, simply increasing the network size with the original data does not improve performance. We show the mean and standard deviation of the final performance averaged over 8 seeds.

| Environment | Behavioral Policy | Baseline | Larger Network | |
|---|---|---|---|---|
| | | | Original Data | SYNTHER |
| halfcheetah-v2 | random | 11.3±0.8 | 11.0±0.7 | **12.8±0.8** |
| | mixed | 44.8±0.7 | 44.8±1.1 | **48.0±0.5** |
| | medium | 48.1±0.2 | 50.2±1.8 | **53.3±0.4** |
| | medexp | 90.8±7.0 | 95.5±5.4 | **100.1±2.7** |
| walker2d-v2 | random | 0.6±0.3 | 2.6±2.1 | **4.3±1.7** |
| | mixed | 85.6±4.6 | 76.4±9.9 | **93.6±3.6** |
| | medium | 82.7±5.5 | 84.5±1.7 | **87.2±1.2** |
| | medexp | 110.0±0.4 | 110.3±0.5 | 110.2±0.3 |
| hopper-v2 | random | 8.6±0.3 | 10.3±5.6 | 19.5±11.2 |
| | mixed | 64.4±24.8 | 62.4±21.6 | **86.8±12.8** |
| | medium | 60.4±4.0 | 61.9±5.9 | 65.1±4.7 |
| | medexp | 101.1±10.5 | 104.6±9.4 | **109.7±4.1** |
| **locomotion average** | | 59.0±4.9 | 59.5±5.5 | **65.9±3.7** |

generates samples that have significantly lower dynamics MSE than explicit augmentations, even for datapoints that are far away from the training data. This implies that a high level of generalization has been achieved by the SYNTHER model, resulting in the ability to generate **novel, diverse, yet dynamically accurate data** that can be used by policies to improve performance.

### 4.1.2 Scaling Network Size

A further benefit we observe from SYNTHER on the TD3+BC algorithm is that upsampled data can enable scaling of the policy and value networks leading to improved performance. As is typical for RL algorithms, TD3+BC uses a small value and policy network with two hidden layers, and width of 256, and a batch size of 256. We consider increasing the size of both networks to be three hidden layers and width 512 (approximately $6\times$ more parameters), and the batch size to 1024 to better make use of the upsampled data in Table 3.

We observe a large overall improvement of **11.7%** for the locomotion datasets when using a larger network with synthetic data (Larger Network + SYNTHER). Notably, when using the original data (Larger Network + Original Data), the larger network performs the same as the baseline. This suggests that the bottleneck in the algorithm lies in the representation capability of the neural network and *synthetic samples from* SYNTHER *enables effective training of the larger network*. This could alleviate the data requirements for scaling laws in reinforcement learning [1, 28]. However, for the IQL and EDAC algorithms, we did not observe an improvement by increasing the network size which suggests that the bottleneck there lies in the data or algorithm rather than the architecture.

### 4.2 Online Evaluation

Next, we show that SYNTHER can effectively upsample an online agent's continually collected experiences. In this section, we follow the sample-efficient RL literature [12, 16] and consider 3 environments from the DeepMind Control Suite (DMC, Tunyasuvunakool et al. [71]) (cheetah-run, quadruped-walk, and reacher-hard) and 3 environments the OpenAI Gym Suite [10] (walker2d, halfcheetah, and hopper). As in Chen et al. [12], D'Oro et al. [16], we choose the base algorithm to be Soft Actor-Critic (SAC, Haarnoja et al. [24]), a popular off-policy entropy-regularized algorithm, and benchmark against a SOTA sample-efficient variant of itself, 'Randomized Ensembled Double Q-Learning' (REDQ, Chen et al. [12]). REDQ uses an ensemble of 10 Q-functions and computes target values across a randomized subset of them during training. By default, SAC uses an update-to-data ratio of 1 (1 update for each transition collected); the modifications to SAC in REDQ enable this to be raised to 20. Our method, 'SAC (SYNTHER)', augments the training data by generating 1M new samples for every 10K real samples collected and samples them with a ratio $r = 0.5$. We then match REDQ and train with a UTD ratio of 20. We evaluate our algorithms over 200K online steps for the DMC environments and 100K for OpenAI Gym.

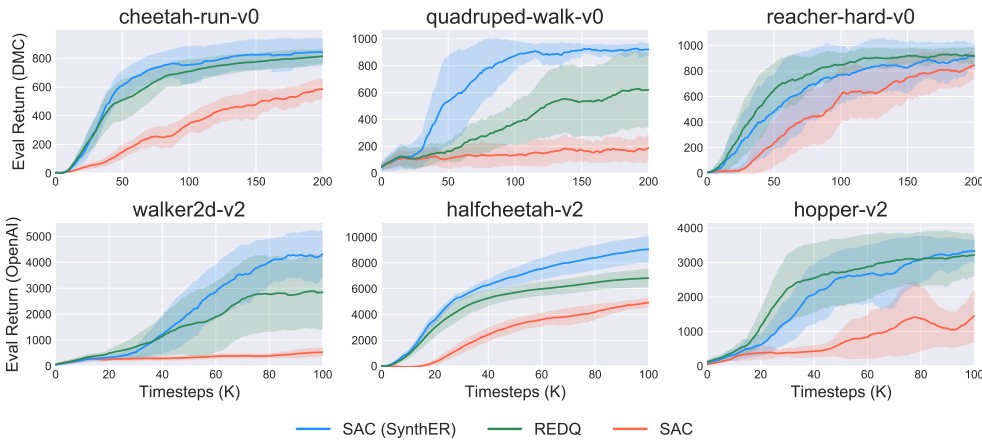

Figure 6: SYNTHER greatly improves the sample efficiency of online RL algorithms by enabling an agent to train on upsampled data. This allows an agent to use an increased update-to-data ratio (UTD=20 compared to 1 for regular SAC) *without any algorithmic changes*. We show the mean and standard deviation of the online return over 6 seeds. DeepMind Control Suite environments are shown in the top row, and OpenAI Gym environments are shown in the bottom.

In Figure 6, we see that SAC (SYNTHER) matches or outperforms REDQ on the majority of the environments with particularly strong results on the quadruped-walk and halfcheetah-v2 environments. This is particularly notable as D'Oro et al. [16] found that UTD=20 on average *decreased performance* for SAC compared with the default value of 1, attributable to issues with overestimation and overfitting [12, 48]. We aggregate the final performance on the environments in Figure 1b, normalizing the DMC returns following Lu et al. [50] and OpenAI returns as in D4RL. Moreover, due to the fast speed of training our diffusion models and fewer Q-networks, our approach is in fact faster than REDQ based on wall-clock time, whilst also requiring fewer algorithmic design choices, such as large ensembles and random subsetting. Full details on run-time are given in Appendix D.2.

### 4.3 Scaling to Pixel-Based Observations

Finally, we show that we can readily scale SYNTHER to pixel-based environments by generating data in the latent space of a CNN encoder. We consider the V-D4RL [50] benchmarking suite, a set of standardized pixel-based offline datasets, and focus on the 'cheetah-run' and 'walker-walk' environments. We use the associated DrQ+BC [50] and BC algorithms. Whilst the original image observations are of size $84 \times 84 \times 3$, we note that the CNN encoder in both algorithms generates features that are 50 dimensional [77]. Therefore, given a frozen encoder pre-trained on the same dataset, we can retain the fast training and sampling speed of our proprioceptive models but now in pixel space. We present full details in Appendix F.

Analogously to the proprioceptive offline evaluation in Section 4.1, we upsample 5M latent transitions for each dataset and present downstream performance in Table 4. Since the V-D4RL datasets are smaller than the D4RL equivalents with a base size of 100K, we would expect synthetic data to be beneficial. Indeed, we observe a statistically significant increase in performance of **+9.5%** and **+6.8%** on DrQ+BC and BC respectively; with particularly strong highlighted results on the medium and expert datasets. We believe this serves as compelling evidence of the scalability of SYNTHER to high-dimensional observation spaces and leave generating data in the original image space, or extending this approach to the online setting for future work.

## 5 Related Work

Whilst generative training data has been explored in reinforcement learning; in general, synthetic data has not previously performed as well as real data on standard RL benchmarks.

**Generative Training Data.** Imre [34], Ludjen [51] considered using VAEs and GANs to generate synthetic data for online reinforcement learning. However, we note that both works failed to match the original performance on simple environments such as CartPole—this is likely due to the use of a

Table 4: We scale SYNTHER to high dimensional pixel-based environments by generating data in the latent space of a CNN encoder pre-trained on the same offline data. Our approach is composable with algorithms that train with data augmentation and leads to a **+9.5%** and **+6.8%** overall gain on DrQ+BC and BC respectively. We show the mean and standard deviation of the final performance averaged over 6 seeds.

| Environment | | DrQ+BC [50] | | BC [50] | |
|---|---|---|---|---|---|
| | | Original | SynthER | Original | SynthER |
| walker-walk | mixed | 28.7±6.9 | 32.3±7.6 | 16.5±4.3 | 12.3±3.6 |
| | medium | 46.8±2.3 | 44.0±2.9 | 40.9±3.1 | 40.3±3.0 |
| | medexp | 86.4±5.6 | 83.4±6.3 | 47.7±3.9 | 45.2±4.5 |
| | expert | 68.4±7.5 | **83.6±7.5** | 91.5±3.9 | 92.0±4.2 |
| cheetah-run | mixed | 44.8±3.6 | 43.8±2.7 | 25.0±3.6 | 27.9±3.4 |
| | medium | 53.0±3.0 | 56.0±1.2 | 51.6±1.4 | 52.2±1.2 |
| | medexp | 50.6±8.2 | 56.9±8.1 | 57.5±6.3 | **69.9±9.5** |
| | expert | 34.5±8.3 | **52.3±7.0** | 67.4±6.8 | **85.4±3.1** |
| Average | | 51.7±5.7 | 56.5±5.4 **(+9.5%)** | 49.8±4.2 | 53.2±4.1 **(+6.8%)** |

weaker class of generative models which we explored in Section 3.1. Huang et al. [32] considered using GAN samples to *pre-train* an RL policy, observing a modest improvement in sample efficiency for CartPole. Chen et al. [13], Yu et al. [79] consider augmenting the image observations of robotic control data using a guided diffusion model whilst maintaining the same action. This differs from our approach which models the entire transition and *can synthesize novel action and reward labels*.

Outside of reinforcement learning, Azizi et al. [7], He et al. [26], Sariyildiz et al. [60] consider generative training data for image classification and pre-training. They also find that synthetic data improves performance for data-scarce settings which are especially prevalent in reinforcement learning. Sehwag et al. [64] consider generative training data to improve adversarial robustness in image classification. In continual learning, "generative replay" [65] has been considered to compress examples from past tasks to prevent forgetting.

**Generative Modeling in RL.** Prior work in diffusion modeling for offline RL has largely sought to supplant traditional reinforcement learning with "upside-down RL" [62]. Diffuser [36] models long sequences of transitions or full episodes and can bias the whole trajectory with guidance towards high reward or a particular goal. It then takes the first action and re-plans by receding horizon control. Decision Diffuser [4] similarly operates at the sequence level but instead uses conditional guidance on rewards and goals. Du et al. [17] present a similar trajectory-based algorithm for visual data. In contrast, SYNTHER operates at the transition level and seeks to be readily compatible with existing reinforcement learning algorithms. Pearce et al. [55] consider a diffusion-based approach to behavioral cloning, whereby a state-conditional diffusion model may be used to sample actions that imitate prior data. Azar et al. [6], Li et al. [47] provide theoretical sample complexity bounds for model-based reinforcement learning given access to a generative model.

**Model-Based Reinforcement Learning.** We note the parallels between our work and model-based reinforcement learning [35, 49, 78]; which tends to generate synthetic samples by rolling out using forward dynamics models. Two key differences of this approach to our method are: SYNTHER synthesizes new experiences without the need to start from a real state and the generated experiences are distributed exactly according to the data, rather than subject to compounding errors due to modeling inaccuracy. Furthermore, SYNTHER is an orthogonal approach which could in fact be *combined with* forward dynamics models by generating initial states using diffusion, which could lead to increased diversity.

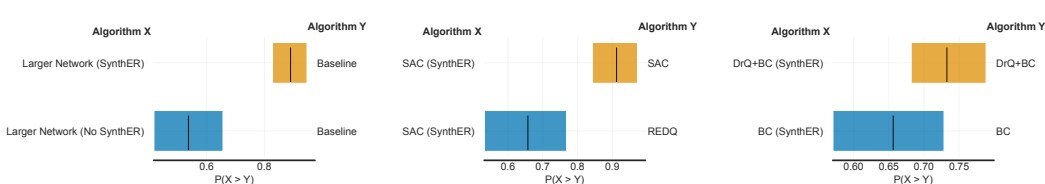

(a) Offline TD3+BC Larger Networks (Full results in Table 3)  (b) Online DMC Evaluation (Full results in Figure 6)  (c) Offline V-D4RL Evaluation (Full results in Table 4)

Figure 7: RLiable [3] analysis allowing us to aggregate results across environments and show the probability of improvement for SYNTHER across our empirical evaluation.

# 6   Conclusion

In this paper, we proposed SYNTHER, a powerful and general method for upsampling agent experiences in any reinforcement learning algorithm using experience replay. We integrated SYNTHER with ease on **six distinct algorithms across proprioceptive and pixel-based environments**, each fine-tuned for its own use case, **with no algorithmic modification**. Our results show the potential of synthetic training data when combined with modern diffusion models. In offline reinforcement learning, SYNTHER allows training from extremely small datasets, scaling up policy and value networks, and high levels of data compression. In online reinforcement learning, the additional data allows agents to use much higher update-to-data ratios leading to increased sample efficiency.

We have demonstrated that SYNTHER is a scalable approach and believe that extending it to more settings would unlock extremely exciting new capabilities for RL agents. SYNTHER could readily be extended to $n$-step formulations of experience replay by simply expanding the input space of the diffusion model. Furthermore, whilst we demonstrated an effective method to generate synthetic data in latent space for pixel-based settings, exciting future work could involve generating transitions in the original image space. In particular, one could consider fine-tuning large pre-trained foundation models [59] and leveraging their generalization capability to synthesize novel views and configurations of a pixel-based environment. Finally, by using guidance for diffusion models [29], the generated synthetic data could be biased towards certain modes, resulting in transferable and composable sampling strategies for RL algorithms.

### Acknowledgments

Cong Lu is funded by the Engineering and Physical Sciences Research Council (EPSRC). Philip Ball is funded through the Willowgrove Studentship. The authors would like to thank the anonymous Reincarnating Reinforcement Learning Workshop at ICLR 2023 and NeurIPS 2023 reviewers for positive and constructive feedback which helped to improve the paper. We would also like to thank Shimon Whiteson, Jakob Foerster, Tim Rocktäschel and Ondrej Bajgar for reviewing earlier versions of this work.

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

# Supplementary Material

## A    Data Modeling

In this section, we provide further details for our data modeling. Our diffusion model generates full environment transitions i.e., a concatenation of states, actions, rewards, next states, and terminals where they are present. For the purposes of modeling, we normalize each continuous dimension (non-terminal) to have 0 mean and 1 std. We visualize the marginal distributions over the state, action, and reward dimensions on the standard halfcheetah medium-replay dataset in Figure 8 and observe that the synthetic samples accurately match the high-level statistics of the original dataset.

We note the difficulties of appropriately modeling the terminal variable which is a binary variable compared to the rest of the dimensions which are continuous for the environments we investigate. This is particularly challenging for "expert" datasets where early termination is rare. For example, walker2d-expert only has $\approx 0.0001\%$ terminals. In practice, we find it sufficient to leave the terminals un-normalized and round them to 0 or 1 by thresholding the continuous diffusion samples in the middle at 0.5. A cleaner treatment of this variable could be achieved by leveraging work on diffusion with categorical variables [31].

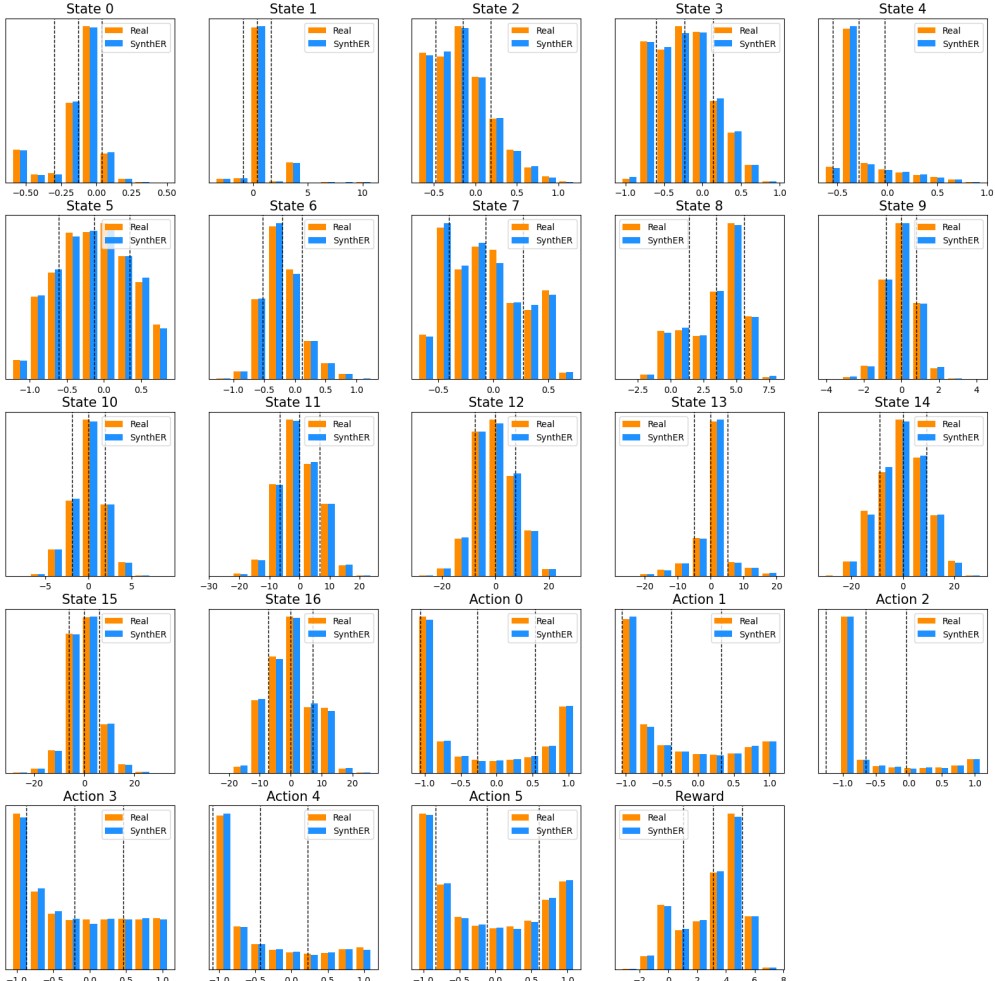

Figure 8:    Histograms of the empirical marginal distribution of samples from SYNTHER in **blue** on the halfcheetah medium-replay dataset against the original data in **orange**. Dashed lines indicate the mean $\pm$ one standard deviation in the original dataset. SYNTHER faithfully reproduces the high-level statistics of the dataset.

## A.1 Data Compression

An immediate advantage of sampling data from a generative model is compression. In Table 5, we compare the memory requirements of SYNTHER and the original data by the number of 32-bit floating point numbers used by each for some sample D4RL [21] datasets. For the original data, this simply scales linearly with the size of the dataset. On other hand, SYNTHER amortizes this in the number of parameters in the denoising network, resulting in a high level of dataset compression, at the cost of sampling speed. This property was also noted in the continual learning literature with generative models summarizing previous tasks [65]. As we discuss in Appendix B.3, sampling is fast with 100K transitions taking around 90 seconds to generate.

Table 5: SYNTHER provides high levels of dataset compression *without sacrificing downstream performance* in offline reinforcement learning. Statistics shown are for the standard D4RL MuJoCo walker2d datasets which has a transition dimension of 42, and the residual denoiser used for evaluation on these environments in Section 4.1. Figures are given to 1 decimal place.

| Dataset | # FP32s in Original Dataset | # Diffusion Parameters | Compression |
|---|---|---|---|
| mixed | 12.6M | | 1.9× |
| medium | 42M | 6.5M | 6.5× |
| medium-expert | 84M | | 12.9× |

# B  Hyperparameters

## B.1  TVAE and CTGAN

In Section 3.1, we compared SYNTHER to the VAE and GAN baselines, TVAE and CTGAN. As these algorithms have not been used for reinforcement learning data before, we performed a hyperparameter search [42] across the following spaces:

Table 6: Hyperparameter search space for TVAE. We highlight the default choice in **bold**.

| Parameter | Search Space |
|---|---|
| no. layers | { 1, **2**, 3, 4 } |
| width | { 64, **128**, 256, 512 } |
| batch size | { 250, **500**, 1000 } |
| embedding dim | { 32, 64, **128**, 256 } |
| loss factor | { 0.02, 0.2, **2**, 20} |

Table 7: Hyperparameter search space for CTGAN. We highlight the default choice in **bold**.

| Parameter | Search Space |
|---|---|
| no. layers | { 1, **2**, 3, 4 } |
| width | { 64, 128, **256**, 512 } |
| batch size | { 250, **500**, 1000 } |
| embedding dim | { 32, 64, **128**, 256 } |
| discriminator steps | { **1**, 2} |

These ranges are similar to those listed in Tables 10 and 11 of Kotelnikov et al. [42]. We used 30 trials along with the default.

## B.2  Denoising Network

The formulation of diffusion we use in our paper is the Elucidated Diffusion Model (EDM, Karras et al. [38]). We parametrize the denoising network $D_\theta$ as an MLP with skip connections from the previous layer as in Tolstikhin et al. [70]. Thus each layer has the form given in Equation (3).

$$x_{L+1} = \text{linear}(\text{activation}(x_L)) + x_L \tag{3}$$

The hyperparameters are listed in Table 8. The noise level of the diffusion process is encoded by a Random Fourier Feature [57] embedding. The base size of the network uses a width of 1024 and

depth of 6 and thus has $\approx$ 6M parameters. We adjust the batch size for training based on dataset size. For online training and offline datasets with fewer than 1 million samples (medium-replay datasets) we use a batch size of 256, and 1024 otherwise.

For the following offline datasets, we observe more performant samples by increasing the width up to 2048: halfcheetah medium-expert, hopper medium, and hopper medium-expert. This raises the network parameters to $\approx$ 25M, which remains fewer parameters than the original data as in Table 5. We provide ablations on the depth and type of network used in Table 10.

Table 8: Default Residual MLP Denoiser Hyperparameters.

| Parameter | Value(s) |
|---|---|
| no. layers | 6 |
| width | 1024 |
| batch size | { 256 for online and medium-replay, 1024 otherwise } |
| RFF dimension | 16 |
| activation | relu |
| optimizer | Adam |
| learning rate | $3 \times 10^{-4}$ |
| learning rate schedule | cosine annealing |
| model training steps | 100K |

### B.3 Elucidated Diffusion Model

For the diffusion sampling process, we use the stochastic SDE sampler of Karras et al. [38] with the default hyperparameters used for the ImageNet, given in Table 9. We use a higher number of diffusion timesteps at 128 for improved sample fidelity. We use the implementation at `https://github.com/lucidrains/denoising-diffusion-pytorch` which is released under an Apache license.

Table 9: Default ImageNet-64 EDM Hyperparameters.

| Parameter | Value |
|---|---|
| no. diffusion steps | 128 |
| $\sigma_{\min}$ | 0.002 |
| $\sigma_{\max}$ | 80 |
| $S_{\text{churn}}$ | 80 |
| $S_{\text{tmin}}$ | 0.05 |
| $S_{\text{tmax}}$ | 50 |
| $S_{\text{noise}}$ | 1.003 |

The diffusion model is fast to train, taking approximately 17 minutes for 100K training steps on a standard V100 GPU. It takes approximately 90 seconds to generate 100K samples with 128 diffusion timesteps.

## C SYNTHER Ablations

We consider ablations on the number of generated samples and type of denoiser used for our offline evaluation in Section 4.1.

### C.1 Size of Upsampled Dataset

In our main offline evaluation in Section 4.1, we upsample each dataset (which has an original size of between 100K to 2M) to 5M. We investigate this choice for the walker medium-replay dataset in Figure 9 and choose 10 levels log-uniformly from the range [50K, 5M]. Similarly to He et al. [26], we find that performance gains with synthetic data eventually saturate and that 5M is a reasonable heuristic for all our offline datasets.

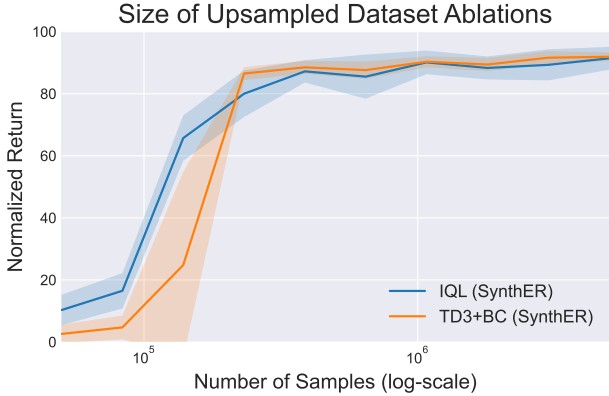

Figure 9: Ablations on the number of samples generated by SYNTHER for the offline walker medium-replay dataset. We choose 10 levels log-uniformly from the range [50K, 5M]. We find that performance eventually saturates at around 5M samples.

## C.2 Network Ablations

We ablate the hyperparameters of the denoising network, comparing 3 settings of depth from $\{2, 4, 6\}$ and analyze the importance of skip connections. The remaining hyperparameters follow Appendix B.2. We choose the hopper medium-expert dataset as it is a large dataset of 2M. As we can see in Table 10, we see a positive benefit from the increased depth and skip connections which leads to our final choice in Table 8.

Table 10: Ablations on the denoiser network used for SYNTHER on the hopper medium-expert dataset. We observe that greater depth and residual connections are beneficial for downstream offline RL performance. We show the mean and standard deviation of the final performance averaged over 4 seeds.

| Network | Depth | Eval. Return |
|---|---|---|
| MLP | 2 | 86.8±18.7 |
| | 4 | 89.9±17.9 |
| | 6 | **100.4± 6.9** |
| Residual MLP | 2 | 78.5±11.3 |
| | 4 | **99.3±14.7** |
| | 6 | **101.1±10.5** |

## D  RL Implementation

For the algorithms in the offline RL evaluation in Section 4.1, we use the 'Clean Offline Reinforcement Learning' (CORL, Tarasov et al. [68]) codebase. We take the final performance they report for the baseline offline evaluation. Their code can be found at `https://github.com/tinkoff-ai/CORL` and is released under an Apache license.

For the online evaluation, we consider Soft Actor-Critic (SAC, Haarnoja et al. [24]) and use the implementation from the REDQ [12] codebase. This may be found at `https://github.com/watchernyu/REDQ` and is released under an MIT license. We use the 'dmcgym' wrapper for the DeepMind Control Suite [71]. This may be found at `https://github.com/ikostrikov/dmcgym` and is released under an MIT license.

### D.1  Data Augmentation Hyperparameters

For the data augmentation schemes we visualize in Figure 1a, we define:

1. Additive Noise [45]: adding $\epsilon \sim \mathcal{N}(0, 0.1)$ to $s_t$ and $s_{t+1}$.
2. Multiplicative Noise [45]: multiplying $s_t$ and $s_{t+1}$ by single number $\epsilon \sim \text{Unif}([0.8, 1.2])$.

3. Dynamics Noise [8]: multiplying the next state delta $s_{t+1} - s_t$ by $\epsilon \sim \text{Unif}([0.5, 1.5])$ so that $s_{t+1} = s_t + \epsilon \cdot (s_{t+1} - s_t)$.

## D.2 Online Running Times

Our online implementation in Section 4.2 uses the default training hyperparameters in Appendix B.2 to train the diffusion model every 10K online steps, and generates 1M transitions each time. On the 200K DMC experiments, 'SAC (SynthER)' takes $\approx 21.1$ hours compared to $\approx 22.7$ hours with REDQ on a V100 GPU. We can further break down the running times of 'SAC (SynthER)' as follows:

- Diffusion training: 4.3 hours
- Diffusion sampling: 5 hours
- RL training: 11.8 hours

Therefore, the majority of training time is from reinforcement learning with an update-to-data ratio (UTD) of 20. We expect the diffusion training may be heavily sped-up with early stopping, and leave this to future work. The default SAC algorithm with UTD=1 takes $\approx 2$ hours.

## E  Further Offline Results

In this section, we include additional supplementary offline experiments to those presented in Section 4.1.

### E.1  AntMaze Data Generation

We further verify that SYNTHER can generate synthetic data for more complex environments such as AntMaze [21]. This environment replaces the 2D ball from Maze2D with the more complex 8-DoF "Ant" quadruped robot, and features: non-Markovian policies, sparse rewards, and multitask data. In Table 11, we see that SYNTHER improves the TD3+BC algorithm where it trains (on the 'umaze' dataset) and achieves parity otherwise.

Table 11: We show synthetic data from SYNTHER achieves at least parity for more complex offline environments like AntMaze-v2, evaluated with the TD3+BC algorithm. We show the mean and standard deviation of the final performance averaged over 6 seeds.

| Environment | | TD3+BC [22] | |
| --- | --- | --- | --- |
| | | Original | SynthER |
| AntMaze | umaze | 70.8±39.2 | **88.9±4.4** |
| | medium-play | 0.3±0.4 | 0.5±0.7 |
| | large-play | 0.0±0.0 | 0.0±0.0 |

### E.2  Offline Data Mixtures

In Section 4.1, we considered exclusively training on synthetic data. We present results in Table 12 with a 50-50 mix of real and synthetic data to confirm that the two are compatible with each other, similar to Section 4.2. We do so by including as many synthetic samples as there are real data. As we stated before, we do not expect an increase in performance here due to the fact that most D4RL datasets are at least 1M in size and are already sufficiently large.

Table 12: We verify that the synthetic data from SYNTHER can be mixed with the real data for offline evaluation. The 50-50 mix achieves parity with the original data, same as the synthetic data. We show the mean and standard deviation of the final performance averaged over 8 seeds.

| Environment | TD3+BC [22] | | | IQL [41] | | |
| --- | --- | --- | --- | --- | --- | --- |
| | Original | SYNTHER | 50-50 | Original | SYNTHER | 50-50 |
| **locomotion average** | 59.0±4.9 | 60.0±5.1 | 59.2±3.7 | 62.1±3.5 | 63.7±3.5 | 62.7±5.1 |

# F  Latent Data Generation with V-D4RL

We provide full details for the experiments in Section 4.3 that scale SYNTHER to pixel-based environments by generating data in latent space for the DrQ+BC [50] and BC algorithms. Concretely, for the DrQ+BC algorithm, we consider parametric networks for the shared CNN encoder, policy, and $Q$-functions, $f_\xi$, $\pi_\phi$, and $Q_\theta$ respectively. We also use a random shifts image augmentation, aug. Therefore, the $Q$-value for a state $s$ and action $a$ is given by $Q_\theta(f_\xi(\text{aug}(s)), a)$. The policy is similarly conditioned on an encoding of an augmented image observation.

The policy and $Q$-functions both consist of an initial 'trunk' which further reduces the dimensionality of the CNN encoding to $d_{\text{feature}} = 50$, followed by fully connected layers. We represent this as $\pi_\phi = \pi_\phi^{\text{fc}} \circ \pi_\phi^{\text{trunk}}$ and $Q_\theta = Q_\theta^{\text{fc}} \circ Q_\theta^{\text{trunk}}$. This allows us to reduce a pixel-based transition to a low-dimensional latent version. Consider a pixel-based transition $(s, a, r, s')$ where $s, s' \in \mathbb{R}^{84 \times 84 \times 3}$. Let $h = f_\xi(\text{aug}(s))$ and $h' = f_\xi(\text{aug}(s'))$. The latent transition we generate is:

$$(\pi_\phi^{\text{trunk}}(h), Q_\theta^{\text{trunk}}(h), a, r, \pi_\phi^{\text{trunk}}(h'), Q_\theta^{\text{trunk}}(h'))$$

This has dimension $4 \cdot d_{\text{feature}} + |a| + 1$ and includes specific supervised features for both the actor and the critic; we analyze this choice in Appendix F.1. For example, on the 'cheetah-run' environment considered in V-D4RL, since $|a| = 6$, the overall dimension is 207 which is suitable for our residual MLP denoising networks using the same hyperparameters in Table 8. This allows us to retain the fast training and sampling speed from the proprioceptive setting but now in pixel space.

To obtain a frozen encoder $f_\xi$ and trunks $\pi_\phi^{\text{trunk}}$, $Q_\theta^{\text{trunk}}$, we simply train in two stages. The first stage trains the original algorithm on the original data. The second stage then retrains only the fully-connected portions of the actor and critic, $\pi_\phi^{\text{fc}}$ and $Q_\theta^{\text{fc}}$, with synthetic data. Thus, our approach could also be viewed as fine-tuning the heads of the networks. The procedure for the BC algorithm works the same but without the critic.

We use the official V-D4RL [50] codebase for the data and algorithms in this evaluation. Their code can be found at `https://github.com/conglu1997/v-d4rl` and is released under an MIT license.

## F.1  Ablations On Representation

We analyze the choice of low-dimensional latent representation we use in the previous section, in particular, using specific supervised features for both the actor and critic. We compare this against using actor-only or critic-only features for both the actor and critic, which corresponds to a choice of $\pi_\phi^{\text{trunk}} = Q_\theta^{\text{trunk}}$, in Table 13. We note that both perform worse with an especially large drop-off for the critic-only features. This may suggest that non-specific options for compressing the image into low-dimensional latents, for example, using auto-encoders [40], could be even less suitable for this task.

Table 13: Ablations on the latent representation used for SYNTHER on the V-D4RL cheetah expert dataset. We observe that separate specific supervised features are essential for downstream performance with a particularly large decrease if we only used critic features for the actor and critic. We show the mean and standard deviation of the final performance averaged over 4 seeds.

| Latent Representation | Eval. Return |
|---|---|
| Actor and Critic (Ours) | **52.3±7.0** |
| Actor Only | 43.5±7.3 |
| Critic Only | 16.0±2.8 |

