# OpenReview forum: "Synthetic Experience Replay"
_NeurIPS.cc/2023/Conference — NeurIPS 2023 poster_

### Official Review · Reviewer_oBNH · 2023-06-18

**Soundness:** 4 excellent
**Presentation:** 4 excellent
**Contribution:** 4 excellent
**Rating:** 7
**Confidence:** 5

**Summary:**

Authors propose to use diffusion models for generating new data based on online interactions or offline dataset. Generating new samples allows online and offline algorithms to perform better. Method can be useed with any offline method and any online method which utilize replay buffers.

**Strengths:**

Useful idea which works. Both offline and online RL settings  are covered as well as both visual and non-visual tasks. Good ablations and large-scale experiments.

**Weaknesses:**

I  would recommend to validate the approach using Antmaze D4RL datasets as this domain is much more challenging than locomotion and maze2d.

**Questions:**

Maybe I missed it, but how would  performance change in offline setting if we used original data with the synthetic one? Does it decrease performance? I don't understand the motivation to throw away original data otherwise except demonstrating that diffusion produces good data which could be an ablation.

**Limitations:**

I'm don't knowvery much about how complicated it is to train a diffusion model to solve an arbitary task so making it work seems like a potential bottleneck. Probably, it is faster and easier to run an additional hyperparameter search for the RL algorithm than make diffusion work.

---

> ### Author Rebuttal · Authors · 2023-08-08
>
> We thank the reviewer for their time to read our paper and provide useful comments. We now address their individual concerns.
>
> **Antmaze experiments**
>
> We thank the reviewer for this comment. We would like to point out that the larger dataset offline experiments are more for validation (e.g., ensuring that diffusion models can faithfully model the distribution such that it doesn’t harm performance). As we noted in our response to reviewer 2GdG, we note that the D4RL datasets are by construction relatively large and generated under stochastic policies and starting points, therefore we’d expect the benefit from SynthER to be limited in this setting as the dataset is already relatively diverse.
>
> Nevertheless, we show results for AntMaze using the TD3+BC algorithm in Table 3 of the supplementary PDF. We verify that diffusion can indeed also match the original data and even improve on the umaze dataset. We hope this provides additional confidence in our approach.
>
> **Mixture experiments**
>
> The reviewer is right, the motivation to throw away the offline data is simply to make the point that the diffusion generated samples are high enough quality to faithfully model the original distribution, in particular when that dataset is large and diverse. However, the reviewer raises an interesting point regarding the mixture of offline and diffused data. In Table 4 of our supplementary PDF, we show that similar to the online experiments, there is no problem with mixing real and synthetic offline data in a 50-50 mix. As we discussed in the paper, the performance with the real, synthetic and mixed data are close to each other.
>
> **Ease of training diffusion**
>
> Thank you for the question, we find that it is actually the exact opposite. The diffusion model uses one single set of hyperparameters for almost all experiments and requires no additional tuning. This is in stark contrast to typical RL hyperparameter sweeps. Furthermore, we unlock performance that is likely not possible even with simply sweeping different hyperparameters on the base RL algorithms.
>
> We once again thank the reviewer for their constructive comments, which have improved the quality of our experiments and clarity of our paper, and kindly ask them to raise their score if they believe we have addressed their concerns. If issues still remain, then we would be more than happy to discuss these.

---

> > ### Comment · Reviewer_oBNH · 2023-08-11
> >
> > Thank you very much for answering my questions and running additional experiments. I'm increasing the confidence score.

---

> > > ### Author Response · Authors · 2023-08-17
> > >
> > > Thank you so much again for your feedback which greatly improved our paper!

---

### Official Review · Reviewer_MwZd · 2023-07-02

**Soundness:** 3 good
**Presentation:** 4 excellent
**Contribution:** 3 good
**Rating:** 7
**Confidence:** 4

**Summary:**

The paper presents Synthetic Experience Replay, a reinforcement learning algorithm employing a generative model based on diffusion to enrich the training dataset of a learning agent. The method is adapted to offline and online RL, and compared to traditional augmentation strategies (e.g., the addition of random noise) in continuous control tasks, from proprioceptive and visual inputs, while applied to different policy optimization algorithms. The paper also presents detailed analyses on the approach, including results on the quality of the generated data and on using larger neural networks.

**Strengths:**

**Originality**: the general direction of using synthetic data for training machine learning algorithms has been vastly explored in the past, with some applications to reinforcement learning. The approach presented in the paper is, thus, not outstandingly novel. However, one of the main points in the paper is that the main bottleneck in previous approaches leveraging synthetic data in the context of RL has been the quality of the generative model. To the best of my knowledge, this is the first use of diffusion models for synthetic data generation.

**Quality**: the quality of the work is reasonably good. The experiments are well-chosen and the experimental designs are generally good.

**Clarity**: the writing quality is generally very high. The paper is well-written and easy to follow in most parts, with a clear presentation of the experimental results.

**Significance**: I believe demonstrating the effectiveness of powerful generative models in generating synthetic data for reinforcement learning algorithms to use is important for the community, given the simplicity and generality of the approach.

**Weaknesses:**

**Major Concerns**
- My main concern is on the number of seeds employed for the experiments in the paper. I understand the authors might be subject to computational constraints, but I find 4 repetitions only to be generally not enough to fully trust an individual experiment. I would suggest the authors to run more seeds per experiment and, if possible, use the methods proposed in "Deep Reinforcement Learning at the Edge of the Statistical Precipice" (Agarwal at al., 2021) for aggregation across multiple tasks.
- In Section 3.1, the paper says that Tabular VAE and Conditional Tabular GAN have been evaluated using the default hyperparameters proposed in the original paper that applies these generative models to tabular data. However, no dataset from the benchmark employed in the original paper contains robotics interactions similar to the one from D4RL, and this thus begs to question of whether better hyperparameters for such different datasets might exist. I believe the paper would be improved by evaluating those approaches after a careful search for optimal hyperparameters.

**Minor Concerns**
- Some potentially important references are missing from the paper. In particular, I'm thinking of the recent paper on using synthetic experiences (meta-learned) "Should Models Be Accurate?" (Saleh et al., 2022), and of some recent papers applying, albeit with other goals, diffusion models to reinforcement learning such as "Planning with Diffusion for Flexible Behavior Synthesis" (Janner et al., 2022) and "Is Conditional Generative Modeling all you need for Decision-Making? (Ajay et al., 2022).
- The clarity of the paper would benefit from a more explicit description of how the diffusion model is employed for generating the data. One can guess it is naively applied to generating each dimension of each state, action or reward, but being explicit is better than being implicit in this case.

**Questions:**

I ask the author to address the concerns highlighted above in my review.

**Limitations:**

The computational limitations of the approach are stated in the paper, saying that the approach is computationally better than REDQ. However, the comparison is executed at a replay ratio of 20; it would be desirable to understand what is the actual computational cost of the new data generation and training of the diffusion model even at lower replay ratios.

---

> ### Author Rebuttal · Authors · 2023-08-08
>
> We thank the reviewer for their time to read our paper and provide useful comments. We now address their individual concerns.
>
> **More seeds**
>
> Thank you for pointing out this oversight on our behalf. We have now increased the seeds used for all the experiments in our paper to at least 6 and show as many as possible in Figure 1 and Table 1 of the supplementary PDF. We have now also performed a meta-analysis over our empirical evaluation using the RLiable framework in Figure 2 of the supplementary PDF. As the reviewer can see, our existing findings still stand, and in fact with the additional seeds as suggested, we achieve statistically significant results over prior methods.
>
> **Tuning of other generative models**
>
> Thank you for pointing out this oversight on our behalf. We have now tuned the TVAE and CTGAN models following the method of [1]. We used a similar extensive search space for the models as listed in Table 10 and Table 11 of the Appendix of [1] and used 30 trials each for the halfcheetah-mixed dataset. The results are presented in Table 2 of our supplementary PDF.
>
> With extensive tuning, we are able to improve the performance of TVAE and CTGAN on the halfcheetah-mixed dataset. However, this still remains roughly half of what is possible with the diffusion model. We will update the final version of the paper with these results.
>
> [1] TabDDPM: Modelling Tabular Data with Diffusion Models. Akim Kotelnikov, Dmitry Baranchuk, Ivan Rubachev, Artem Babenko.
>
> **How was the data generated?**
>
> The reviewer is absolutely correct in assuming we generate all states/actions/rewards/next states at once (e.g. joint diffusion). We discuss this in Appendix A but will include this point more explicitly in the camera ready version.
>
> **Breakdown of online training time**
>
> Thank you for the question, in Appendix D.1 we list the running time of the “SAC (SynthER)” runs as 21.1 hours. We can further break this down into
> Diffusion training: 4.3 hours
> Diffusion sampling: 5 hours
> RL training: 11.8 hours
> Therefore, the majority of training time is from reinforcement learning with an update-to-data ratio (UTD) of 20. We will include this breakdown in the camera ready.
>
> **Citations**
>
> We appreciate these pointers to references, in fact, we already cite Ajay et al. (2022) and Janner et al. (2022) in our paper (please see citation 3 and 35). However, we appreciate the pointer to Saleh et al. (2022) and will aim to discuss the paper in the camera ready.
>
> We once again thank the reviewer for their constructive comments, which have improved the clarity and thoroughness of our paper, and kindly ask them to raise their score if they believe we have addressed their concerns. If issues still remain, then we would be more than happy to discuss these.

---

> > ### Comment · Reviewer_MwZd · 2023-08-10
> > **Thank you for your work on improving the paper**
> >
> > Dear authors,
> >
> > Good job! Thank you for running this additional experiments and for your answer. My main concerns were addressed and I believe the rigor of the experimentation has been much improved by the new results. Please make sure to put the rliable plots in the final version of the paper.
> >
> > I will raise my score to 7.

---

> > > ### Author Response · Authors · 2023-08-17
> > >
> > > Thank you so much again for your feedback which greatly improved our paper!

---

### Official Review · Reviewer_2GdG · 2023-07-06

**Soundness:** 4 excellent
**Presentation:** 4 excellent
**Contribution:** 4 excellent
**Rating:** 7
**Confidence:** 4

**Summary:**

The paper proposes Synthetic Experience Replay (SYNTHER), a novel diffusion-based approach to upsample an agent's collected experience in deep reinforcement learning (RL). The authors demonstrate that SYNTHER is effective for training RL agents in both offline and online settings, and can improve performance for small offline datasets and larger networks. The paper presents results in both state-based and pixel-based environments and provides evidence that synthetic training data can enable more sample-efficient and scalable training for RL agents.


**Strengths:**

1. The paper introduces a novel approach, SYNTHER, which employ generative diffusion model to upsample an agent's collected experience. This method can potentially resolves the challenge of data scarcity in RL. Generally, this method is easy to understand and effective, and may have a broad impact on sample-efficient RL algorithms.
2. The authors provide thorough experiments and evaluations in offline and online settings, demonstrating the effectiveness of SYNTHER in improving performance and sample efficiency. The authors also demonstrated the superiority of the diffusion-based model compared to other generative models such as GAN and VAE.
3. The paper is well-structured, with clear introduction, background and methodology. The author also provide open-source code for reproducibility.

**Weaknesses:**

1. The performance improvement of SYNTHER seems to be very limited under the setting of small networks.
2. The paper didn't demonstrate the advantages of using SYNTHER on existing RL algorithms in both online and pixel based scenarios, nor did it study the impact on model-based RL algorithms in online settings.

**Questions:**

1. Can you provide the experimental results of SYNTHER with a model-based RL method without BC?
2. Can you provide the experimental results of SYNTHER in online and pixel-based setting?
3. SYNTHER can only generate 1-step transition, so n-step bootstrap or GAE cannot be used in imagined data. Did the baselines use n-step bootstrap or GAE value estimation?

**Limitations:**

SYNTHER have limited results in pixel-based and online settings. This paper has no negative social impacts.

---

> ### Author Rebuttal · Authors · 2023-08-08
>
> We thank the reviewer for their time to read our paper and provide useful comments. We now address their individual concerns.
>
> **The role of the offline experiments/small networks**
>
> We thank the reviewer for this point; we note that the D4RL datasets are by construction relatively large and generated under stochastic policies and starting points, therefore we’d expect the benefit from SynthER to be limited in this setting as the dataset is already relatively diverse. We also note small networks more or less can solve for these environments in the default dataset setting. Finally, we view the full offline experiments as more validation that SynthER can scale to larger datasets while still faithfully capture their underlying distribution so as not to harm behavior learning.
>
> **Model-based**
>
> We see model-based as being relatively orthogonal to our work. While SynthER could of course be deployed alongside a model-based method, it’s unclear how best to use the diffusion data, as model-based methods already produce their own “synthetic” data through imagined rollouts; do we run these two processes independently or somehow combine them? One immediate direction could be to improve the quality of a world model using SynthER, but we note that the key challenge in RL lies more with issues of over-fitting in the TD-learning loss function, and regularization is required to carefully handle this [1,2]; in this work we regularize using diffusion-based data augmentation. We would expect to see limited benefits on augmenting the already ‘stable’ supervised signal in world model learning, and moreover would be disinclined to add additional complexity and training time to the already demanding model-based RL framework.
>
> [1] Stabilizing Off-Policy Deep Reinforcement Learning from Pixels, Cetin et al., ICML 2022.
>
> [2] Efficient Deep Reinforcement Learning Requires Regulating Overfitting, Li et al., ICLR 2023.
>
> **Online pixel-based**
>
> This is an interesting research direction, which we believe is out of scope for the purposes of this paper. A key issue is addressing multi-modality to generating high-dimensional images alongside low-dimensional actions and rewards. Whilst we present some results in the paper combining the proprioceptive architecture with a fixed latent representation, we believe this is not optimal in general for pixel-based applications, particularly as learned latent representations are likely to change over the course of training.
>
> We tried some experiments using intermediate representations from Convolutional U-Nets to jointly diffuse actions/rewards alongside images, but initial results were not promising, particularly for modeling the low-dimensional action/reward distribution. This suggests a more complex approach is required. Given that multimodal diffusion is a significant on-going body of unresolved research, we therefore leave pixel-based experiments to future work.
>
> We once again thank the reviewer for their constructive and interesting comments, which have improved the clarity of our paper, and kindly ask them to raise their score if they believe we have addressed their concerns. If issues still remain, then we would be more than happy to discuss these.

---

> > ### Comment · Reviewer_2GdG · 2023-08-19
> >
> > Thanks for your response. However, the last question was not effectively answered. I will keep my score at 7.

---

> > > ### Author Response · Authors · 2023-08-19
> > > **Additional Response**
> > >
> > > Thank you so much for your response. We apologize that we missed the reviewer's final question, as it was not in the original review. We will now answer this here.
> > >
> > > **SYNTHER can only generate 1-step transition, so n-step bootstrap or GAE cannot be used in imagined data. Did the baselines use n-step bootstrap or GAE value estimation?**
> > >
> > > This statement is untrue, SynthER can be extended to $n$-step transitions by expanding the input to the diffusion model. We refer the reviewer to Line 305 of our manuscript which addresses this. Concretely, instead of modeling 1-step transitions by diffusion tuples of the form $(s, a, r, s')$, we could model expanded tuples of the form $(s, a, r, s', a', r', s'')$ and so on. This represents only a modest increase in the dimensionality of the input space and can easily be handled by the diffusion model. This would allow us to use $n$-step bootstrap or GAE methods.
> > >
> > > The proprioceptive results used 1-step bootstraps. The latent offline visual experiments in fact already use a 3-step bootstrap with a latent that is derived from a 3-step transition.
> > >
> > > Please let us know if this has answered the reviewer's doubt or if we can elaborate further.

---

### Official Review · Reviewer_jt3y · 2023-07-06

**Soundness:** 3 good
**Presentation:** 3 good
**Contribution:** 2 fair
**Rating:** 6
**Confidence:** 5

**Summary:**

Building on the recent success of generative models, the authors propose Synthetic Experience Replay to improve how experience or data is used within reinforcement learning algorithms. While data is normally collected through interaction with an environment, the authors suggest that the experience replay can be augmented with additional data now artificially generated from a diffusion model. The paper considers both the online and offline settings of RL, compares with previous tabular data augmentation methods, and measures performance on pixel-based environments.

**Strengths:**

+ [Quality] The authors conduct an exhaustive set of experiments on standard online and offline RL benchmarks.
+ [Clarity] The paper is well-motivated, organized, and clearly written. Related works are carefully documented and contributions are clear.
+ [Significance] The results are quite compelling, especially in the case of upsampling from small datasets. Considering that no algorithmic changes are necessary, many works stand to benefit from this simple, yet effective strategy.

**Weaknesses:**

+ [Originality] Augmenting data in the replay buffer with synthetic transitions is not new. For example, previous works have attempted to learn a dynamics model during the course of standard online RL training to fill up the replay buffer with fake transitions. Most of the novelty in this work comes from 1) using a diffusion model to generate data and 2) demonstrating experimentally that this kind of synthetic data can be useful in a variety of settings.

**Questions:**

-

**Limitations:**

-

---

> ### Author Rebuttal · Authors · 2023-08-08
>
> We thank the reviewer for their time to read our paper and provide useful comments. We now address their individual concerns.
>
> **Novelty**
>
> Thank you for the question - we appreciate the reviewer’s concerns. While we agree that other methods have been proposed to generate synthetic data, the strength of our approach is in it’s simplicity. Indeed, several highly influential recent RL papers, published at similar venues, have shown that simple ideas can be both intuitive and highly effective [1,2,3].
>
> On this point, we note that not only do we demonstrate the capacity for diffusion models to faithfully model the distribution of a replay buffer (unlike prior classes of generative models), but we also observe its utility in augmenting and expanding effective buffer size, allowing us to address issues with overfitting that can arise in RL [4,5]. This results in significant improvement over existing approaches, despite surprisingly relying on minimal underlying algorithmic modification (i.e., just train our diffusion model on the replay buffer). To speak to this point about superlative results, we observe a statistically significant improvement of SynthER over prior methods in the new experiments we ran through the RLiable framework (see Figure 2 in the supplementary PDF in the general response).
>
> [1] A Minimalist Approach to Offline Reinforcement Learning, Fujimoto and Gu, NeurIPS 2021.
>
> [2] Image Augmentation Is All You Need: Regularizing Deep Reinforcement Learning from Pixels, Kostrikov et al., ICLR 2021.
>
> [3] Mastering Visual Continuous Control: Improved Data-Augmented Reinforcement Learning, Yarats et al., ICLR 2022.
>
> [4] Randomized Ensembled Double Q-Learning: Learning Fast Without a Model, Chen et al., ICLR 2021.
>
> [5] Efficient Deep Reinforcement Learning Requires Regulating Overfitting, Li et al., ICLR 2023.
>
> We once again thank the reviewer for their constructive and interesting comments, which have improved the clarity of our paper, and kindly ask them to raise their score if they believe we have addressed their concerns. If issues still remain, then we would be more than happy to discuss these.

---

> > ### Author Response · Authors · 2023-08-19
> >
> > We would like to thank the reviewer again for their time in reviewing our manuscript. As the discussion period is coming to a close, we were wondering if the reviewer had any further questions or queries that we could address.
> >
> > Thanks again!

---

### Author Rebuttal · Authors · 2023-08-08

We would like to thank the reviewers for their helpful and valuable feedback. We were pleased that the reviewers found our paper well-written, believed that our idea was widely applicable, and agreed that our experiments were thorough and of a high standard.

We noticed that each reviewer had fairly unique concerns, but observed a shared theme regarding the need for additional experimental validation. We are excited to share improvements to our experiments in our supplementary PDF and will discuss overall improvements in the general response. Any remaining individual concerns will be addressed in the specific reviewer responses.

**More seeds**

Thank you to reviewer MwZd for raising this, we have now increased the number of seeds used in all the experiments to a minimum of 6 each. We show as many as possible in Figure 1 and Table 1 of the supplementary PDF. We see no significant change in results, which should increase confidence in the robustness and reproducibility of our experiments.

**RLiable analysis**

As additional validation, we evaluated SynthER under the RLiable framework in Figure 2 of the supplementary PDF, where we observe statistical significance in SynthER’s improvements over the baselines.

Finally, we thank all the reviewers for their constructive and interesting comments, which have improved the clarity and experimental rigor of our paper. We take this opportunity to therefore kindly ask them to raise their score if they believe we have addressed their concerns; if issues still remain, then we would be more than happy to discuss these in the coming days.

---

### Decision · Program_Chairs · 2023-09-21

**Decision:**

Accept (poster)

**Comment:**

This paper proposes to use diffusion models to generate synthetic data for replay-based RL algorithms. The authors show that the proposed method is effective in both offline and online RL settings.

The authors addressed all the concerns of the reviewers and reviewers increased their scores and confidence based on the rebuttal. I recommend an acceptance. I strongly recommend the authors to update the final version of the paper with more seeds as shown in the rebuttal pdf.